# Identification and characterization of piperine synthase from black pepper, *Piper nigrum* L.

Arianne Schnabel [1], Benedikt Athmer[1], Kerstin Manke[1], Frank Schumacher[2], Fernando Cotinguiba [3] &
Thomas Vogt [1 ✉]

Black pepper (*Piper nigrum* L.) is the world's most popular spice and is also used as an ingredient in traditional medicine. Its pungent perception is due to the interaction of its major compound, piperine (1-piperoyl-piperidine) with the human TRPV-1 or vanilloid receptor. We now identify the hitherto concealed enzymatic formation of piperine from piperoyl coenzyme A and piperidine based on a differential RNA-Seq approach from developing black pepper fruits. This enzyme is described as piperine synthase (piperoyl-CoA:piperidine piperoyl transferase) and is a member of the BAHD-type of acyltransferases encoded by a gene that is preferentially expressed in immature fruits. A second BAHD-type enzyme, also highly expressed in immature black pepper fruits, has a rather promiscuous substrate specificity, combining diverse CoA-esters with aliphatic and aromatic amines with similar efficiencies, and was termed piperamide synthase. Recombinant piperine and piperamide synthases are members of a small gene family in black pepper. They can be used to facilitate the microbial production of a broad range of medicinally relevant aliphatic and aromatic piperamides based on a wide array of CoA-donors and amine-derived acceptors, offering widespread applications.

[1] Leibniz Institute of Plant Biochemistry, Dept. Cell and Metabolic Biology, Halle (Saale), Germany. [2] Core Facility Vienna Botanical Gardens, Vienna, Austria. [3] Instituto de Pesquisas de Produtos Naturais (IPPN), Universidade Federal do Rio de Janeiro (UFRJ), Rio de Janeiro/RJ, Brasil. ✉email: tvogt@ipb-halle.de

The Piperaceae are tropical and subtropical plants within the Piperales, a large order of the Magnoliids, showing properties of basal angiosperms[1]. Dried fruits of several Piperaceae, specifically black pepper (*Piper nigrum*) have been used as popular spices by humans since antiquity, and, in the 15th and 16th century were among the driving economic forces leading to the discovery of the *New World*. Besides flavor, black pepper fruits show a wide range of applications in traditional and modern medicine[2–5]. The pungent perception of black pepper is largely due to high concentrations of several amides, specifically piperine (1-piperoyl-piperidine), which is regarded as the basis for traditional and recent therapeutic applications[6,7]. Piperine results in an oral burning sensation due to activation of the transient receptor cation channel subfamily V member 1 (TRPV-1), formerly known as the vanilloid receptor[8]. This ion channel is also targeted by capsaicin, a structurally similar compound from *Capsicum* species (hot chili peppers), that are members of the Solanaceae[9].

Piperine was isolated 200 years ago by Hans Christian Ørstedt[10] and numerous procedures for organic synthesis of piperamides are constantly developed[11]. Yet, the biosynthetic steps towards piperine and piperamide formation in black pepper until recently have remained largely enigmatic (Fig. 1). Early reports on the incorporation of *L*-lysine and cadaverine into the piperidine heterocycle by radiolabeled tracers date back to five decades ago and were performed at that time with Crassulaceae species, rather than black pepper[12]. The vanilloid-like aromatic part of piperine and its structural similarity to ferulic acid suggested that its extended C5-carbon side chain may be derived from the general phenylpropanoid pathway, although experimental evidence for this claim is rather poor. Feeding studies with 2-[$^{13}$C]-and 2-[$^{2}$H]-labeled malonic acid as well as $^{15}$N-labeled L-valine suggested the participation of a CoA-activated malonyl coenzyme A and valine into the similar isobutylamine derived piperlongumine in *Piper tuberculatum*[13]. Several piperoyl-CoA ligases capable of converting piperic acid to piperoyl-coenzyme A (piperoyl-CoA) have now been described from black pepper immature fruits and leaves, respectively[14,15]. A cytochrome P450 oxidoreductase (CYP719A37) was identified in parallel from immature black pepper fruits[16]. The enzyme catalyzes methylenedioxy bridge formation, specifically from feruperic acid to piperic acid, and not from ferulic acid or from feruperine (Fig. 1). These recent reports corroborate earlier assumptions that amide formation is the final step in piperine biosynthesis[17]. A corresponding piperine synthase activity (EC 2.3.1.145) from crude protein extracts of black pepper shoots capable to convert piperoyl-CoA and piperidine to piperine was already reported three decades ago[17]. Since the presumably similar amide forming capsaicin synthase that is encoded by the *PUN1* (*Pungency 1*) locus[18,19] was classified as a coenzyme A dependent BAHD-type acyltransferase termed by the initials of the first four characterized transferases of this family, BEAT, AHCT, HCBT, and DAT[20], a similar type of enzyme may catalyze the formation of piperine.

Next-generation sequencing technologies enable the assembly of whole transcriptomes and facilitate the identification of genes and enzymes from unknown or only partly solved biosynthetic pathways in non-model organisms[21–23]. Several RNA-Seq-based transcriptome datasets from mature fruits, leaves, and roots were described from black pepper[24–27]. In addition, genome information from black pepper recently suggested a series of piperamide biosynthesis candidate genes and transcripts, yet without any functional characterization[27]. By a differential RNA-Seq approach we now demonstrate that a specific acyltransferase, termed piperine synthase, isolated from immature black pepper fruits catalyzes the decisive step in the formation of piperine from

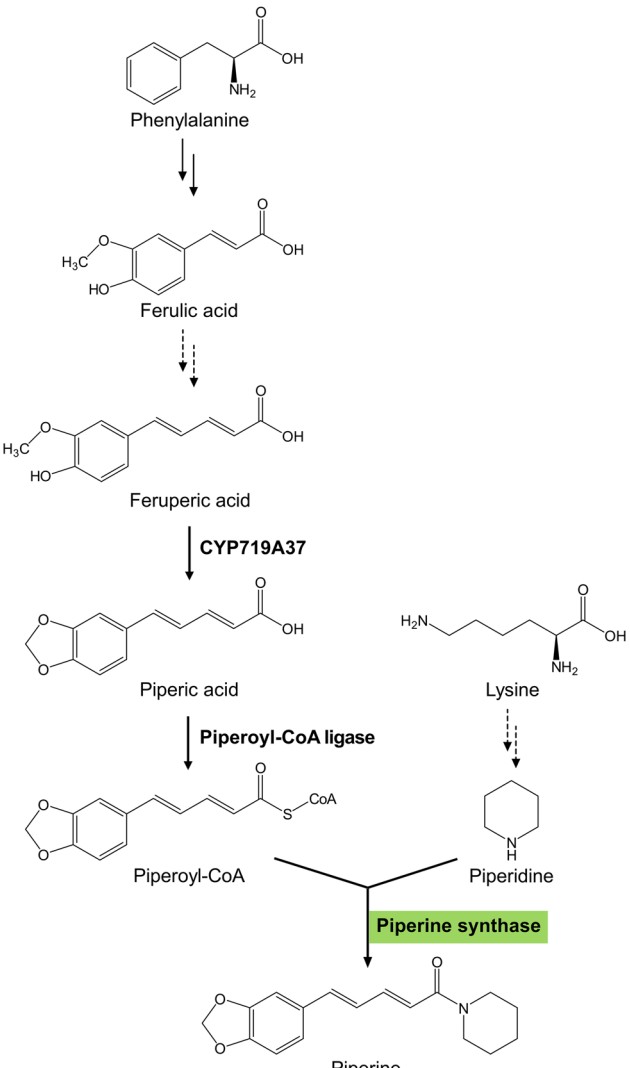

**Fig. 1 Partly hypothetical pathway of piperine biosynthesis in black pepper fruits.** The aromatic part of piperine is presumably derived from the phenylpropanoid pathway, whereas the formation of the piperidine heterocycle appears synthesized from the amino acid lysine. Double and dashed arrows mark either several or unknown enzymatic steps, respectively. Recombinant CYP719A37 and piperoyl-CoA ligase catalyze steps from feruperic acid to piperic acid and to piperoyl-CoA subsequently[15,16]. Piperine synthase, identified and functionally characterized in this report, is highlighted in gray and catalyzes the terminal formation of piperine from piperidine and piperoyl-CoA.

piperoyl-CoA and piperidine. This identification was based on the assumption that piperine synthase is differentially expressed in fruits, leaves, and flowers, with the highest expression levels anticipated for young fruits. Piperine synthase is dependent on activated CoA-esters[14] and therefore, is part of the BAHD-superfamily of acyltransferases[20,28].

## Results

### RNA-sequencing and bioinformatics guided identification of piperine biosynthesis genes. To identify piperine biosynthesis-related genes we monitored piperine formation during fruit development of black pepper plants grown in a greenhouse over several months (Fig. 2a, b). Spadices of individual plants were marked and piperine amounts were quantified by LC-MS and UV/Vis-detection respectively (Fig. 2b). A time course showed

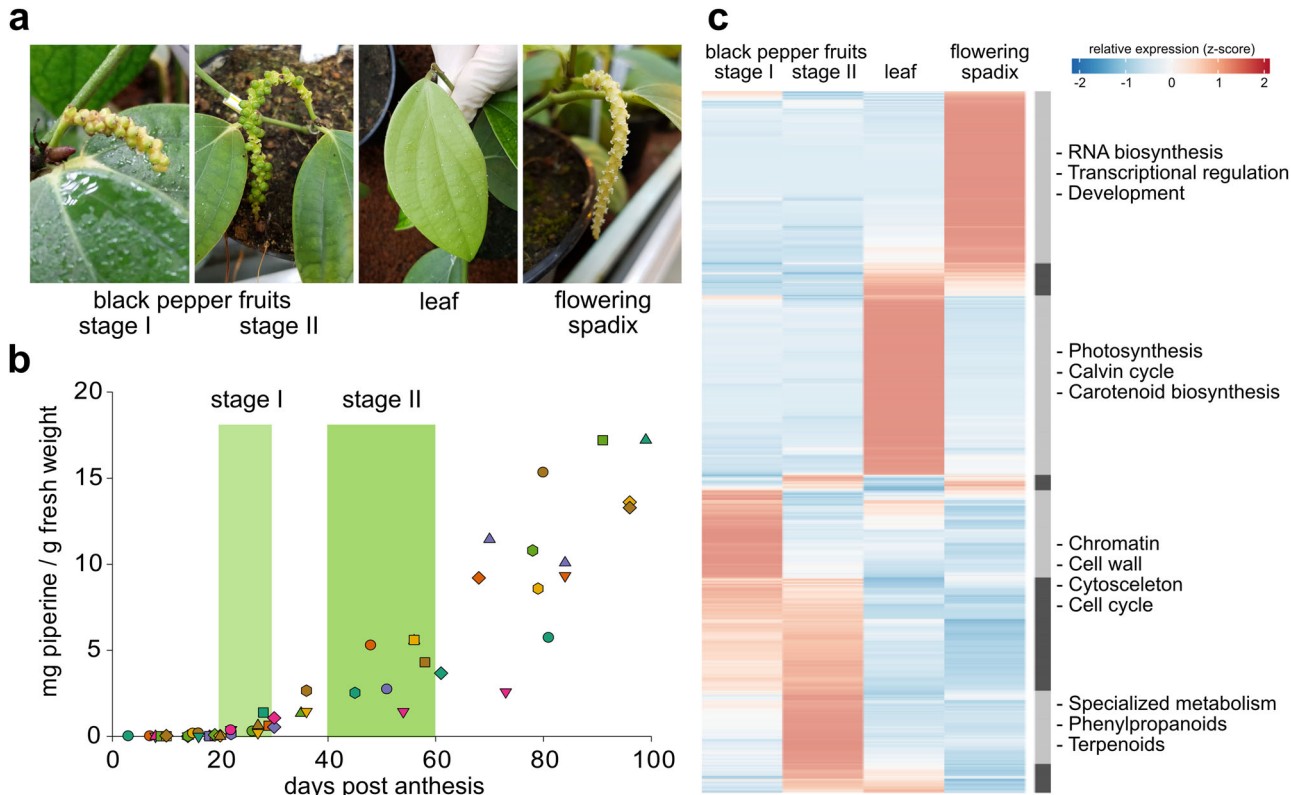

**Fig. 2 Screening for piperine biosynthesis-related genes. a** Illustration of different black pepper organs selected for the RNA-Seq data approach.
**b** Piperine accumulation over 100 days of fruit development. Stages I (20–30 days) and II (40–60 days) are marked in (light) green boxes. Each dot marks
the piperine content of a single fruit picked from different spadices at a certain time. **c** Heatmap of the top differentially expressed genes and functional
annotation. Three thousand most significant differentially expressed genes of each statistical comparison (false discovery rate (FDR) < 0.2, |LFC| > 1) were
used as an input for HOPACH hybrid clustering. Gene set analysis was performed on "first level" clusters and over-represented categories (FDR < 0.001)
were exemplified and highlighted. RNA-Seq data were generated from individual organs in three biological replicates.

that piperine accumulation in greenhouse-grown plants started
after a lag-phase of roughly 20 days post anthesis and peaked
~3 months post anthesis at levels of 2.5% piperine calculated per
fresh weight. No significant increase was observed during later
stages of fruit development. Two development stages, between 20
and 30 days (stage I) and between 40 and 60 days post anthesis
(stage II), were considered as ideal for collecting plant material
for a comparative RNA-Seq approach assuming that transcript
levels preceded the accumulation of biosynthetic product for-
mation. Flowering spadices as well as young leaves served as
negative controls. In both of these organs, either no or very low
piperine amounts were detected by LC-MS analysis. The results of
the RNA-Seq based differential screening approach are illustrated
in Fig. 2c, supported by Supplementary Fig. S1. The first two
principal components (PC) demonstrate high variability between
tissue samples and high reproducibility of the datasets since
samples from biological replicates cluster together (Supplemen-
tary Fig. S1). PC 1 represents a large proportion (94%) of the
variability in the data set and divides the samples into fruit and
non-fruit samples. The heat map generated with the 3000 most
differentially expressed unigenes indicates that individual organs
show unique transcript profiles. Leaf samples are highly abundant
in photosynthesis-related transcripts, while flowering spadices
harbor a diverse mix of developmentally and regulatory tran-
scripts. Gene set analysis (GSEA) of the fruit stage I specific
cluster revealed over-representations with a false discovery rate
(FDR) < 0.001 related to chromatin, cell cycle, cytoskeleton, and
cell wall organization emphasizing the ongoing cell expansion in
this early stage of fruit development.

The MapMan4-based protein classification[29] of the com-
bined transcriptomes resulted in a high score for unigenes
related to an "enzyme classification" bin (Supplementary
Fig. S1). Specifically, the transition from fruit stage I to fruit
stage II marks an over-representation of transcripts linked to
enzymes of plant specialized metabolism, accentuating the
initiation of terpenoid and phenylpropanoid biosynthesis.
Abundant transcripts comprise terpene synthases, CoA- ligases,
oxidoreductases, and several acyltransferases, the latter being
potentially capable to catalyze CoA-dependent amide formation
(Supplementary Table S1).

In parallel to the slope of piperine formation, expression
profiles of two highly expressed acyltransferase candidate genes
followed a pattern of highest abundance of piperine in fruits,
stage II, versus fruits at stage I, leaves, and flowering spadices.
Both are listed as number two and number five among the 30
most abundant transcripts associated to plant specialized
metabolism, topped only by a terpene synthase 10-like annotation
(Supplementary Table S1). The transcripts were classified as
transferases with the highest sequence identity to hexen-1-ol-
acetyltransferase and benzoyl-benzoate acyltransferases, respec-
tively. Transcript abundance was further confirmed by RT-qPCR
with higher expression levels of fruits at stage II (40–60 days post
anthesis) as compared to fruits at stage I (20–30 days post
anthesis) (Fig. 3). Transcript levels of both acyltransferases were
virtually absent in leaves and flowering spadices. Minor transcript
levels can be observed in roots, consistent with detectable levels of
piperine in this organ (Fig. 3c). In contrast to fruits, where
piperine levels just start to accumulate and rise to much higher

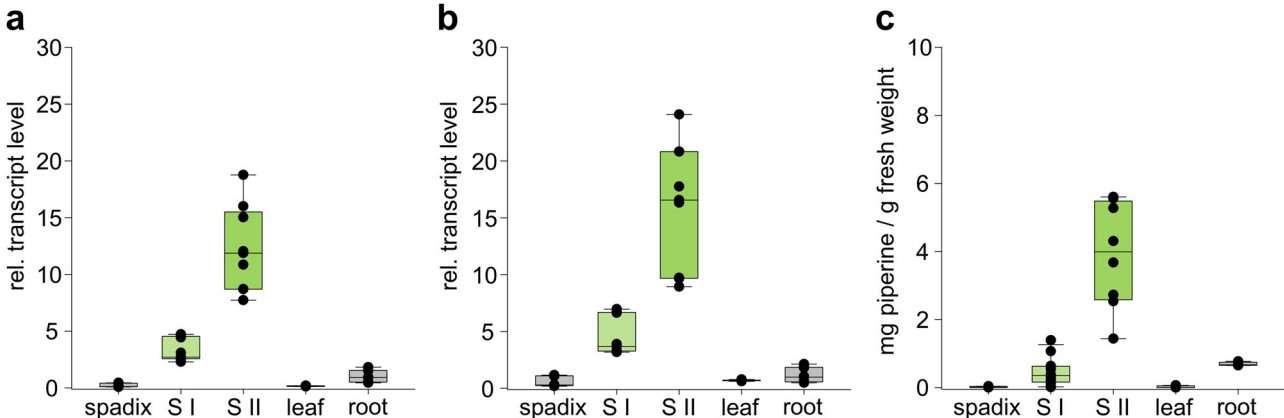

**Fig. 3 Organ-specific relative transcript levels of two black pepper BAHD-type transferases based on RT-qPCR. a** PipBAHD1 (piperamide synthase). **b** PipBAHD2 (piperine synthase). Each box represents nine data points consisting of three biological replicates with three technical replicates each. Transcript levels and also piperine levels of flowering spadices; S I (fruits, stage I, 20–30 days post anthesis); S II (fruits, stage II, 40–60 days post anthesis), young leaves; 5, roots are shown. Elongation factor (elF2B) was used as a reference[15]. **c** Level of piperine in spadices, fruits (stages I and II, 20–30 and 40–60 days post anthesis, respectively), leaves, and roots of black pepper. Piperine levels rise further until ~100 days post anthesis (see also Fig. 1).

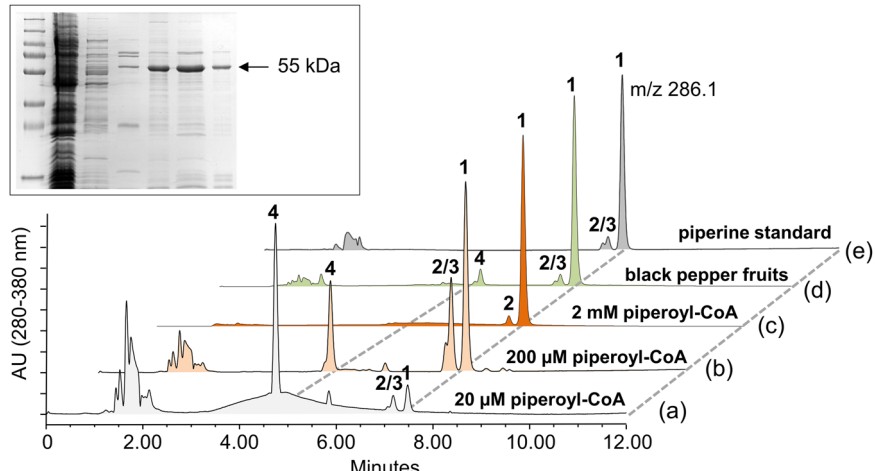

**Fig. 4 Enzyme activity of recombinant and native piperine synthase.** Representative chromatograms (detection by UV/Vis) displaying the formation of piperine (1) at different substrate concentrations of piperoyl-CoA with piperine synthase (**a**–**c**), partly purified native piperine synthase activity (**d**), piperine standard (**e**), (2/3), isopiperine and chavicine, (4), piperic acid. SDS-PAGE of recombinant piperine synthase purification is shown as an insert. The position of the 55 kDa signal representing recombinant piperine synthase during purification is marked with an arrow.

levels in mature fruits (see Fig. 2), root tissue was already collected at a mature level.

**Functional characterization of piperine and piperamide synthases.** Full-length unigenes of both candidates were assembled from sequence reads[30], cloned, and sequenced. Assembled sequences showed expected similarities to BAHD-type acyltransferases. The corresponding full-length cDNAs encoded proteins of 461 and 462 amino acids, with calculated molecular masses of 51 kDa in each case. The deduced protein sequences were named PipBAHD1 and PipBAHD2. Full-length cDNAs were functionally expressed in *E. coli*. After purification by immobilized metal affinity chromatography (IMAC), recombinant His-tagged PipBAHD1 and PipBAHD2 displayed bands around 55 kDa when separated by SDS-PAGE (Fig. 4; Supplementary Fig. S2). This slightly higher than anticipated molecular mass on SDS-PAGE is in line with previous observations of other BAHD-type acyltransferases[28]. Enzyme activities were tested with piperoyl-CoA and piperidine as substrates by HPLC-UV/Vis as

well as HPLC-MS in positive ionization mode (Fig. 4). Only purified PipBAHD2 catalyzed the efficient formation of piperine from piperoyl-CoA and piperidine (Fig. 4). We therefore termed this enzyme piperine synthase. No additional co-factors were required and a pH optimum was recorded at pH 8.0. At low µM substrate concentrations, piperine formation was poor and prominent signals of piperic acid (*m/z* 219) as well as two piperine isomers[31], with identical masses, but slightly different UV-absorbance maxima compared to piperine of 345 and 330 nm were recorded (Fig. 4). The enzyme appears stable for several weeks at −80 °C and over repeated freeze-thaw cycles. DTT was included to reduce enzyme aggregation, resulting in inactive enzymes during size inclusion chromatography (Fig. S3).

When concentrations of piperidine and piperoyl-CoA were gradually increased, preferentially piperine was produced and levels of piperic acid and residual piperine isomers were reduced in parallel (Fig. 4). The activity of recombinant piperine synthase was also compared to piperine formation from crude protein preparations of young black pepper fruits 40–60 days post anthesis. A partially purified native enzyme mix, under the same

substrate concentration again showed the virtually exclusive formation of piperine, traces of stereoisomers and again, only minor levels of piperic acid (Fig. 4).

Based on classical Michaelis-Menten kinetics, recombinant piperine synthase showed an apparent $K_m$ of $342 \pm 60\,\mu M$ for piperoyl-CoA and $7.6 \pm 0.5\,mM$ for piperidine with a calculated $k_{cat}$ of $1.01 \pm 0.16\,s^{-1}$ based on total product formation, i.e., formation of piperine and piperine isomers. Preferred piperine formation at high piperoyl-CoA concentrations is consistent with the observed presence of piperine, the 2E,4E-isomer in fresh mature and also in dried fruits. The lack of any configurational isomerism, usually observed rapidly, when piperine preparations are kept in aqueous solutions indicates a highly coordinated piperine biosynthesis and storage in vivo (Fig. 4 and Supplementary Fig. S2). Piperine synthase shows a preference, but is not specific for piperoyl-CoA and piperidine. Pyrrolidine and isobutylamine were also taken as acceptors of the activated acyl-ester although with 40 and 15% activity as compared to piperidine (Fig. 5). The corresponding mass and UV-signals of both amides, at $m/z$ 272 and $m/z$ 274 respectively, were also detected in minor quantities in commercially available dried peppercorns (Supplementary Fig. S4). 3,4-methylenedioxycinnamoyl-CoA is accepted as an alternative CoA-donor, yet with reduced catalytic activities.

In the case of PipBAHD1 also purified as a His-tagged protein, different piperine isomers were produced with exactly the same substrate preparations of piperoyl-CoA and piperidine (see Supplementary Fig. S2). Product peaks showed identical masses, yet different UV-absorbance spectra and retention times, and piperine were usually synthesized as a side product only, regardless of substrate concentration. Kinetic constants with piperoyl-CoA and piperidine are comparable to piperine synthase (Table 1). The unusual product formation observed for this enzyme is inconsistent with the product profile of fruits from greenhouse-grown plants and also that of dried peppercorns (Supplementary Fig. S4). Therefore, its biological relevance remains obscure and piperoyl-CoA and piperidine may not be its actual in vivo substrates. These observations are in line with higher promiscuity of PipBAHD1 for the CoA-donor and the amine acceptor compared to PipBAHD2, the actual piperine synthase. In contrast to piperine synthase, significant product formation was achieved with medium-chain aliphatic CoA-esters, hexanoyl- and octanoyl-CoA, and residual activity was also observed with benzoyl-CoA and piperidine as well as benzylamine. Piperine synthase was inactive with both substrate combinations (Fig. 5). Therefore, the broad substrate specificity of this enzyme led us to name it piperamide synthase. Piperamide synthase produced a large number of amides with a broad range of applications. Products included aromatic and aliphatic alkamides, with diverse therapeutic, pharmacological and protective properties, covered by several, recent patent applications (e.g., PCT/US2018/054554) describing the potential application of alkamides to treat allergic diseases and pain.

**Sequence analysis and comparison to BAHD-type acyltransferases.** Piperine and piperamide synthases share 62% amino acid sequence identity to each other. They contain the two characteristic HXXXD and the DFGWG motifs characteristic for all BAHD-like enzymes[32] (Supplementary Fig. S5). The histidine and aspartate in the HXXXD motif are conserved as part of the catalytically active site. The usual DFGWG-motif which was claimed to be essential for binding of the CoA-SH cofactor is replaced by a DWGWG motif. This C-terminal motif appears unique among all BAHD-type sequences identified up to now but is located outside of the active site and appears to play a more

general role in the conformation of this type of enzymes[33]. Among hundreds of uncharacterized putative BAHD-like sequences identified on the basis of these motifs (https://blast.ncbi.nlm.nih.gov/Blast.cgi), two enzymatically characterized protein sequences show the highest sequence identity of 42% on the amino acid level to piperine synthase (Fig. 6). The enzyme identified from *Clarkia breweri* flowers, benzoyl benzoate transferase (BBT) is able to catalyze the formation of various volatile benzoyl-esters from benzoyl-CoA and a series of medium-chain aromatic (benzyl and cinnamyl) or aliphatic (geraniol and Z-3-hexen-1-ol) alcohols[28]. The enzyme described from Arabidopsis leaves showed a similar specificity for aliphatic alcohols, but instead of benzoyl-CoA used acetyl-CoA as acyl donor. Distantly similar sequences with unknown substrates clustering within this clade V of the BAHD family are spread throughout the plant kingdom, including basal angiosperms *Amborella trichopoda*, *Nymphaea colorata*, and *Nelumbo nucifera* (sacred lotus). Their specificity remains to be established. Less than 20% sequence identity is observed to capsaicin synthase[17] as well as crystallized and/or functionally characterized vinorine synthase from *Rauwolfia serpentina*, anthocyanin malonyltransferase from *Chrysanthemum morifolium*, and cocaine synthase from *Erythroxylon coca*[33–35].

In summary, based on the matchless substrate and product profile, the low sequence similarities to other BAHDs, and the singular DWGWG motif we suggest that the piperine and piperamide synthases are distinct from all other BAHD-type acyltransferases. Additional black pepper transcripts encoding BAHD-like enzymes, fairly highly expressed also in fruits (Supplementary Table S1) point to a small black pepper acyltransferase gene family that was observed recently also in the black pepper genome[27]. This small gene family may encode a set of different enzymes with potentially overlapping specificities resulting in a blend of aliphatic and aromatic amides in various black pepper organs.

## Discussion

The identification of the two major biosynthetic branches of piperine biosynthetic genes remained enigmatic for several decades, with the exception of scattered labeling studies performed to unravel piperidine heterocycle biosynthesis in Crassulaceae and a single report on the identification of a piperine synthase activity in shoots of black pepper, which was unstable and could not be further characterized[12,17]. The low commercial value of pure piperine and its high abundance in black pepper may have initially contributed to the rather modest attention to decipher the biosynthesis of this universal symbol of spiciness as compared to pharmacologically more relevant indole or isoquinoline alkaloids[36–38]. The limited availability of flowering and fruiting black pepper plants further impaired efforts to investigate piperine biosynthesis in this highly recalcitrant species[13].

We identified the key enzymatic reaction linking the aromatic branch of piperine biosynthesis to the presumably lysine-derived formation of the piperidine heterocycle. Identification of piperine synthase as a BAHD-type acyltransferase came along with a series of puzzling observations. Formation of piperic acid by piperine synthase is the predominant reaction at low substrate concentrations in vitro and, most likely is the direct consequence of the proposed reaction mechanism of BAHD-type acyltransferases[32] where the histidine in the HXXXD motif acts as a catalytic base and in the case of piperic acid formation abstracts a proton from water rather than piperidine. The reactive hydroxyl group then attacks the carbonyl group of piperoyl-CoA as a nucleophile resulting in the hydrolysis of CoA-SH and piperic acid. Precise channeling of metabolites in a hypothetical

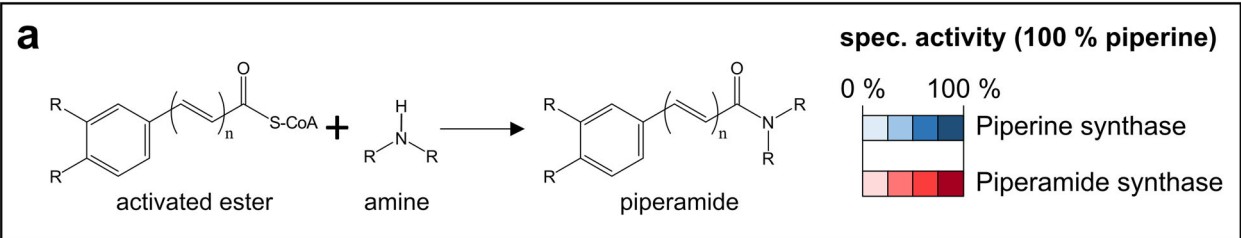

## b product profile of aromatic CoA-donors

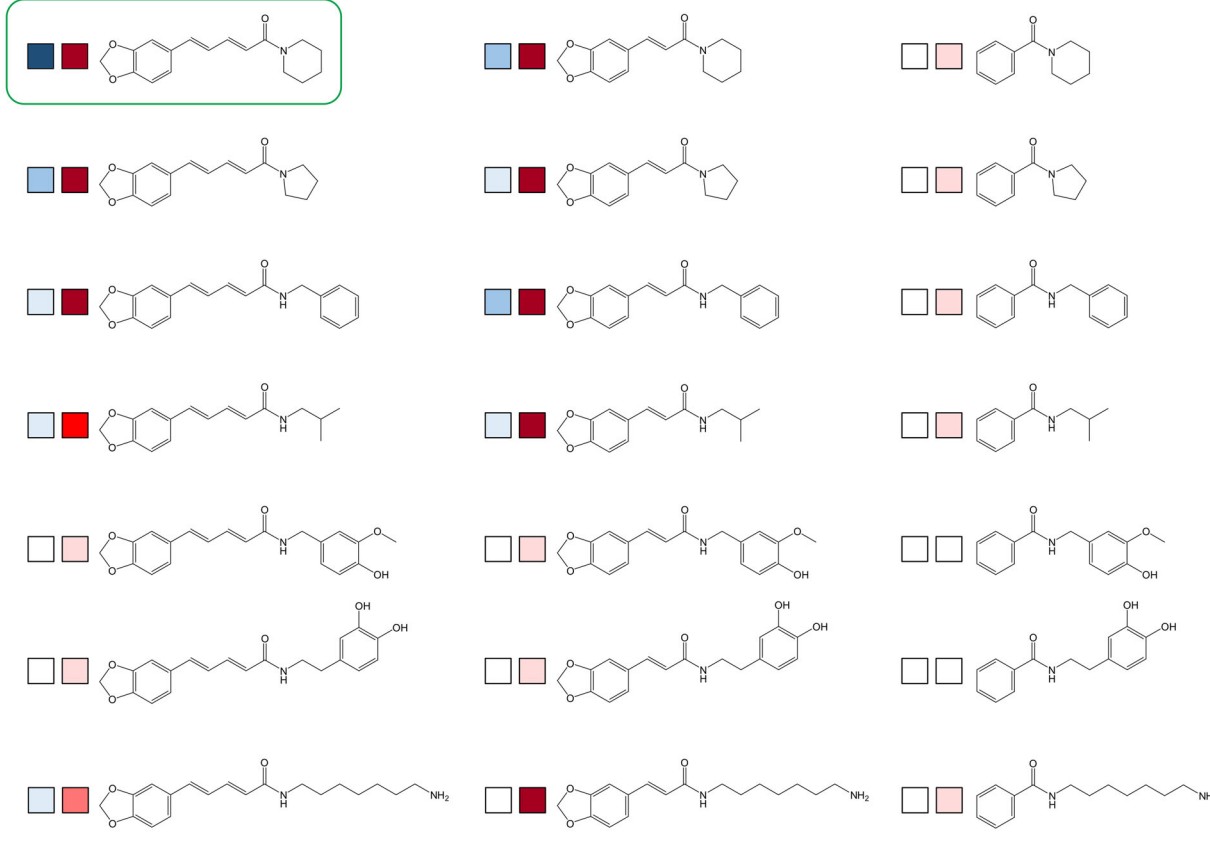

## c product profile of aliphatic CoA-donors

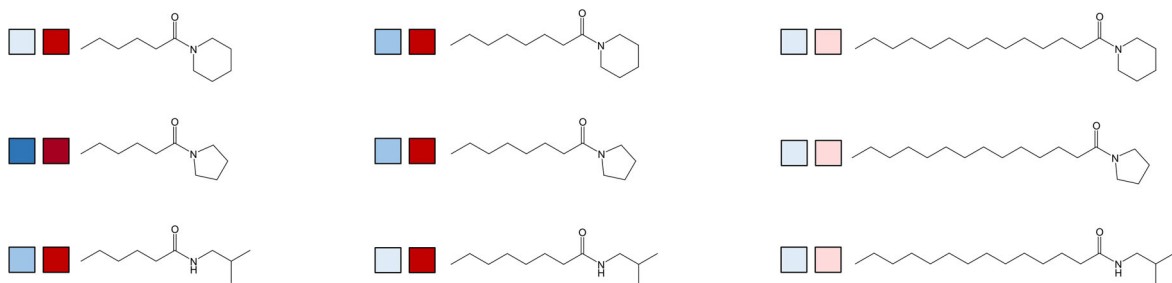

**Fig. 5 Metabolic grid of a large set of amides produced by piperine synthase (blue) and piperamide synthase (red). a** Color intensity of catalyzed reaction is correlated to relative enzyme activity based on LC-UV/Vis and LC-ESI-MS intensities. Relative activities are shown compared to piperine formation, set to 100%, shades of blue piperine synthase, shades of red piperamide synthase. Due to the lack of available standards, and individual, different ionization intensities, relative rather than absolute values can currently be provided with confidence. The combination of CoA-esters and different amines resulted in the production of a large array of amides. **b** Aromatic amides; **c** Aliphatic amides. Piperine synthase appears quite specific for piperoyl-CoA and piperidine, whereas piperamide synthase is substrate promiscuous and tolerates a variety of amines and CoA-esters.

metabolon of piperine formation may reduce the risk of hydrolysis of the CoA-ester in vivo[39–41]. In addition, the specific piperoyl-CoA ligase highly expressed in immature fruits[15] with a low $K_m$ for piperic acid may serve as an anaplerotic enzyme and

theoretically could replenish the pool of piperoyl-CoA at low substrate supply, reducing the accumulation of free piperic acid in the fruits. Piperine synthase is encoded by a single enzyme in the diploid black pepper genome. It is a member of a small gene family which is differentially expressed throughout individual organs[27]. A sequence with 99% sequence identity to the piperine synthase sequence from our transcriptome data was identified from the black pepper genome dataset[27]. In the genome, the gene is neither clustered within any other BAHD-gene nor localized close to potential and tentative pathway candidates, like a lysine/ornithine decarboxylase or the piperoyl-CoA ligase.

The role of piperamide synthase with a preference for the production of alternative stereoisomers from the identical 2E,4E-piperoyl-CoA substrate as compared to piperine synthase remains baffling. Only piperine, rather than its isomers show up in freshly extracted peppercorns, although piperine gradually isomerizes in aqueous solutions[31]. Since the gene encoding piperamide synthase is highly expressed in the fruits we currently assume, that it either produces the "correct" 2E,4E-piperine isomer in vivo or, based on efficient product formation with the substrates

**Table 1 Apparent kinetic constants of piperine synthase and piperamide synthase using piperoyl-CoA (5–2000 μM) and piperidine (1–100 mM) as substrates.**

|  | Apparent $K_m$ [mM] | Apparent $K_{cat}$ [s⁻¹] | Kcat/Km [s⁻¹ M⁻¹] |
|---|---|---|---|
| **Piperine synthase** |  |  |  |
| Piperoyl-CoA | 0.342 ± 0.060 | 1.01 ± 0.12 | 2953 |
| Piperidine | 7.6 ± 0.5 | 0.47 ± 0.11 | 16.2 |
| **Piperamide synthase** |  |  |  |
| Piperoyl-CoA | 0.196 ± 0.009 | 0.35 ± 0.01 | 1786 |
| Piperidine | 8.69 ± 3.6 | 0.27 ± 0.02 | 31.1 |

All data were generated from three individual measurements, performed in triplicates.

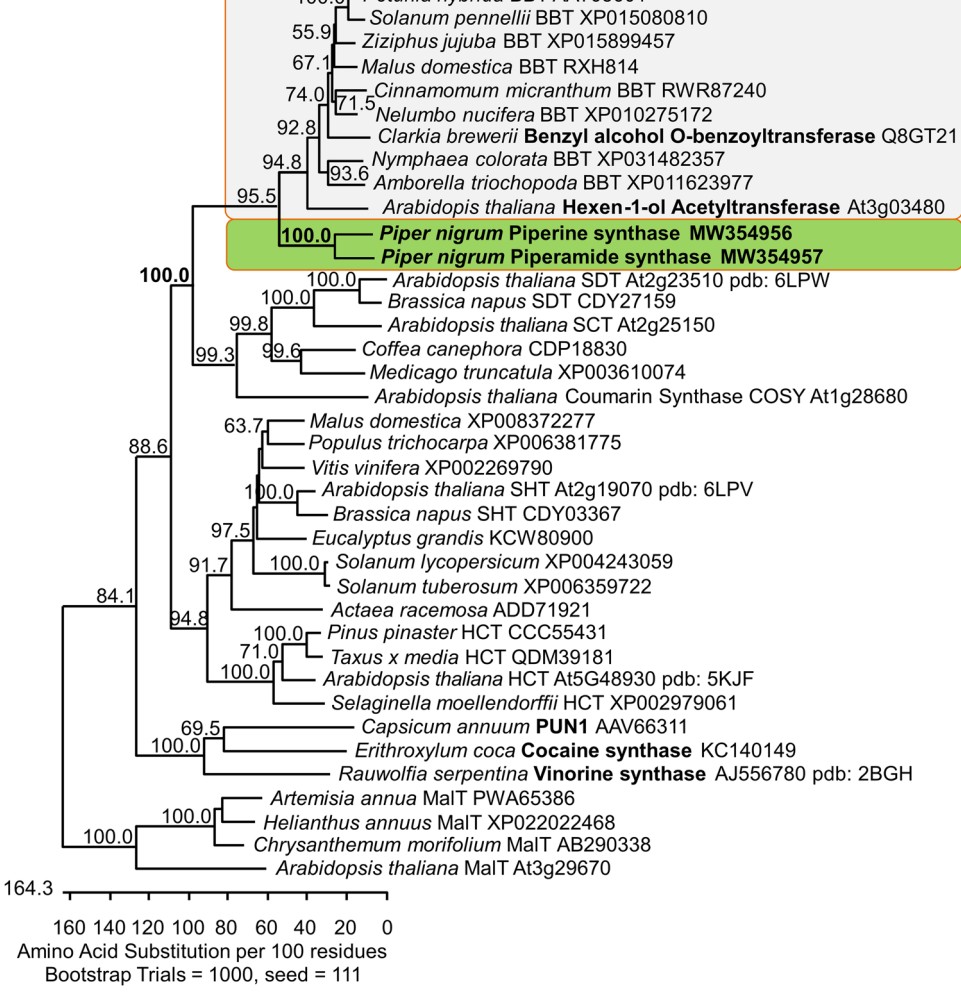

**Fig. 6 Bootstrapped, unrooted cladogram of piperine and piperamide synthases among BAHD-like acyltransferases.** Piperine synthase and closely related piperamide synthase from black pepper are marked in bold green. Functionally characterized benzoyl-CoA benzoate transferases (BBTs) and (Z)-hexen-1-ol acetyltransferase from Arabidopsis (in a gray box)[28,32] represent clade V of BAHD-like acyltransferases but are distinct from piperine and piperamide synthases. Clusters harboring capsaicin synthase[19], vinorine synthase[33], malonoyltransferases[34], cocaine synthase[35], hydroxycinnamoyltransferases (HCTs), and spermidine hydroxycinnamoyl transferases (SHTs, SDTs and SCT) share <25% amino acid sequence identity to piperine and piperamide synthase, respectively. Either NCBI accession numbers, Arabidopsis gene entries (www.tair.org), and/or the pdb-accession number of crystallized and functionally characterized synthases are shown and are also listed in Supplementary Data 1.

hexanoyl- and octanoyl-CoA, respectively, is involved in the synthesis of medium and long-chain aliphatic amides, including (2E,4E)-N-isobutyl-decadienamide or (2E,4E,12Z)-N-isobutyl-octadecatrienamide, isolated from black pepper extracts[42]. The unusual isomer formation in the case of piperine is strikingly similar to an observation recently reported for Arabidopsis coumarin synthase (COSY)[43]. This BAHD-like acyltransferase catalyzes the intramolecular acyl transfer and lactonization, after cis-trans isomerization of 6-ortho-hydroxy-trans-feruloyl-CoA to 6-ortho-hydroxy-cis-feruloyl-CoA, and subsequently to scopoletin in Arabidopsis roots. This isomerization was previously considered to occur spontaneously. A similar type of isomerization appears plausible for piperamide synthase, but requires detailed structural information, one of the immediate goals for future work.

Piperine and related amides are hydrophobic but stored at high concentrations in maturing and mature fruits, raising the question how this can be achieved. The preferred localization of piperine specifically in the perisperm i.e. the seed appears in line with respect to the distribution of these fruits by frugivorous bats or birds, which consume the outer pericarp and leave the perisperm largely undigested[44,45]. There are several scenarios that provide a plausible explanation for the presumably high product concentrations in certain cells. The co-expression of piperine and piperamide synthase with the gene encoding CYP719A37[16], required for methylenedioxy bridge formation indicates the coordinate expression of all pathway genes in a metabolon[41]. The resulting compounds could be stored in small lipid droplets, in differentiated plastids, and in ER-derived microcompartments within the cell or in vesicles and storage devices outside in the apoplast, surrounded by membranes[46]. Since piperine may freely pass these membranes, it remains a mystery how these compounds can be stored without leaking into the cellular symplast. The presence of natural deep eutectic solvents (NADES) provide an intriguing alternative storage possibility for lipophilic, water-insoluble compounds like piperine at high concentrations in specific compartments within a mixture of sugars, proline, and organic acids, e.g., malic acid[47]. Under these conditions enzyme activities can also be conserved, even if water is largely or completely excluded[48,49]. This kind of liquid crystal solubilization may also explain why only a single piperine isomer is detected in dried black peppercorns, whereas in aqueous, methanolic solutions rapid isomerization occurs.

The identification of piperine and corresponding piperamide synthases by a combination of transcriptome and now also of genome data will offer the possibility to synthesize and develop piperine and piperamide analogs by controlled fermentation in heterologous systems[50] rather than organic synthesis, design enzymes with desired properties to be used as catalysts, and engineer the complete piperine biosynthetic pathway into heterologous microbial or eukaryotic hosts. A more detailed structural analysis of both black pepper enzymes will facilitate the design of these new catalysts.

## Methods

**Plant material**. Black pepper (Piper nigrum) cuttings were obtained from the Botanical Garden of the University of Vienna (Austria) from plants collected in 1992 by Dr. R. Samuel, Sri Lanka, IPN No. LK-0-WU-0014181. Plantlets were grown under greenhouse conditions as described previously[15]. Plant material was harvested, ground by a ball mill (Retsch) and stored at −80 °C.

**Preparation of RNA and RNA-Seq analysis**. Total RNA from three biological replicates was isolated from 20–30 d (stage I) and 40–60 d (stage II) old black pepper fruits, young leaves, flowering spadices with NucleoSpin® RNA Plus (Macherey-Nagel) using twice the volume of lysis buffer and binding buffer according to the manufacturer's instructions. Total RNA was quantified by a Nanodrop UV/Vis spectrophotometer (Thermofisher, Dreieich, Germany) and

quality controlled using a QIAxcel capillary electrophoresis system and software (Qiagen, Hilden, Germany). mRNA-sequencing (including library preparation using an unstranded protocol, paired-end sequencing with 150 cycles per read, and demultiplexing of raw data) was carried out by GATC Biotech (Konstanz, Germany). At the time of data generation (2017) no black pepper reference genome[27] was available. Therefore, Illumina HiSeq2000 sequencing was performed and yielded, on average 25 million paired-end reads per sample. Sequencing adapters of reads were removed by Cutadapt and read were subsequently quality trimmed by Trimmomatic. Trinity de novo assembled the cleaned reads into 208.308 genes and 540.916 transcripts, of which 9000 could be classified as full length and annotated by BLAST searches against the curated Swiss-Prot database (https://www.uniprot.org/). Hierarchical clustering of sample distances confirmed high correlation (spearman correlation >0.95) between replicated groups. Sequence data were annotated using the Trinotate and BLAST2GO annotation suites (https://www.blast2go.com/). CAP3 was used to join individual candidate contigs (overlap 200, identity 99%) to obtain full-length transcript sequences. Gene expression analysis was carried out by Trinity which employs BOWTIE2 and RSEM for short read alignment and transcript quantification, respectively. Differential gene expression analysis was performed with edgeR's exactTest using a |log2 fold-change (LFC)| > 1 threshold and a FDR < 0.001. All further data mining and statistical analysis were performed in R (Version 3.6.2). GSEA was performed on the results obtained from HOPACH clustering by using the 3000 most differently expressed genes (FDR < 0.001, |LFC| > 1). The plant-specific MapMan4 functional BIN system was used as input ontology for the cluster-wise gene set testing by clusterProfiler[28].

For RT-qPCR analysis, 200 ng of total RNA and oligo(dT)18-primers were used for cDNA synthesis with Maxima H Minus First Strand cDNA synthesis Kit (Thermo Scientific). The cDNA was diluted 1:10 with water. RT-PCR was run with 3 µl cDNA and 2 pmol of each primer in a 10 µl reaction using qPCR Mix EvaGreen® No Rox (Bio&Sell GmbH) and monitored by CFX Connect Real-Time System (Bio-Rad Laboratories, Inc.). The reference gene P. nigrum elongation factor 2B (eIF2B) was described by us earlier to be fairly expressed in flowering spadices, fruits, leaves, and also in roots[15,16]. All RT-PCRs were performed at least in three biological and individual technical triplicates. A list of all primers is shown in Supplementary Table S2.

**Cloning and enzyme purification**. Total RNA was transcribed with Maxima H Minus First Strand cDNA synthesis Kit (Thermo Scientific) according to manufactures' instructions. Genes were amplified by gene-specific primers with Phusion DNA Polymerase (Thermo Scientific) (Supplementary Table S2). GoTaq G2 Flexi DNA Polymerase (Promega) was used for A-tailing and the resulting product inserted into pGEM-T Easy plasmid (Promega) by T4 DNA Ligase (Promega) and sequenced. After transformation into E. coli DH10B (Thermo Scientific), positive transformants were selected on LB-agar supplemented with 50 µg ml⁻¹ ampicillin. Plasmid purification was performed with NucleoSpin® Plasmid EasyPure (Macherey-Nagel). After digestion with NdeI and BamHI (Thermo Scientific) the genes were inserted in frame into BamHi/NdeI site of pET-16b expression vector (Merck, Darmstadt, Germany), transformed into E. coli LEMO 21 cells (New England Biolabs, Frankfurt, Germany) and selected on 50 µg ml⁻¹ ampicillin and 30 µg ml⁻¹ chloramphenicol. The resulting genes contained an N-terminal His10-Tag for purification by IMAC.

**Protein purification and enzyme assays**. For recombinant protein purification, a pre-culture of 25 ml LB-media containing 50 µg ml⁻¹ ampicillin and 30 µg ml⁻¹ chloramphenicol was inoculated with a single bacteria colony and shaken at 37 °C overnight. A 250-ml liquid culture containing both antibiotics and additional 0.2 mM rhamnose was then inoculated with 5 ml of the pre-culture and shaken 200 rpm at 37 °C. At a cell density of $OD_{600} = 0.7$ the culture was induced by the addition of 1 mM IPTG and shaken for 12–14 h at 25 °C. Cultured cells were pooled and harvested by centrifugation at $10,000 \times g$ for 10 min at 4 °C. Pellets were re-suspended in 50 ml buffer (20 mM Tris/HCl pH 7.5, 100 mM NaCl, 15% glycerol, Buffer A) L⁻¹ of culture and treated with a 10:1 mix of lysozyme and DNaseI 10 mg L⁻¹. Cells were disrupted by ultrasonication, centrifuged at $10,000 \times g$ for 10 min, and to the supernatant protamine sulfate was slowly added to a final concentration of 0.05% to reduce viscosity and centrifuged for 5 min at $20,000 \times g$. The supernatant was applied to a 5 ml Ni-NTA prepacked IMAC-column (Macherey-Nagel). His-tagged recombinant protein was eluted with 400 mM imidazole in Buffer A, fractions were pooled, desalted on a PD-10 column (GE Healthcare), concentrated by a Millipore Ultra-4 Centrifugal Filter Unit (MWCO 10 kDa), directly used for enzyme assays, or stored at −80 °C until use. Determination of the molecular mass was performed by a Sephadex 200 Increase 10/300 column (GE Healthcare) with a buffer containing 20 mM Tris/HCl 8.0, 150 mM NaCl, 1 mM DTT, and 5% glycerol. Protein purification was checked on 12% SDS-PAGE gels and proteins stained by Coomassie Brilliant Blue G 250 (Serva, Heidelberg, Germany). PageRuler© prestained protein ladder (Thermofisher) was used as a size marker. Protein concentrations were calculated based on the corresponding molar extinction coefficients of both recombinant proteins at $\lambda_{280nm}$ using Protean (DNA Star, Madison, Wi, U.S.A.). Total protein yields ranged from <1 mg L⁻¹ in the case of piperine synthase up to 5 mg L⁻¹ fermentation broth for piperamide synthase (Supplementary Fig. S2). Native piperine synthase from 20 g of stage II immature fruits was partially purified as described previously[17].

Piperoyl-CoA was synthesized enzymatically by piperoyl-CoA ligase as previously described from coenzyme A and piperic acid[15]. 3,4-methylenedioxycinnamoyl-CoA was prepared as described in the case of the chemical synthesis of piperoyl-CoA[15]. All other amines, coenzyme A, piperic acid, and CoA-esters used in enzyme assays were commercially available from Merck. All standards were dissolved in aqueous 25–50% DMSO solutions and kept in dark tubes to reduce isomerization at −20 °C until further use. Routinely, 1–2 μg of the purified enzyme was incubated with 200 μM activated ester, 4 mM amine, 30 mM Tris/HCl pH 8.0 and 1 mM DTT in a final volume of 50 μl for up to 30 min. DTT was included to prevent the formation of inactive oligomers, observed during enzyme purification by size exclusion chromatography (Supplementary Fig. S3). A reaction mix without an enzyme to detect and monitor spontaneous amide formation was incubated for the same time and acts as an additional control. Reactions were stopped by the addition of 10 μl of a mixture 50% ACN/10% formic acid (v/v), centrifuged to precipitate protein, and analyzed by reversed-phase HPLC. Piperine formation was analyzed on a 12.5 cm C8 reverse-phase Nucleosil column (Macherey-Nagel) at a flow rate of 0.8 mL min$^{-1}$ and a gradient from 70% aqueous 0.1% formic acid (solvent A) and 30% ACN (solvent B) to 90% solvent B in 10 min. Depending on the substrate and product analyzed, a 5 cm Nucleoshell C18 reverse-phase column was used at a flow rate of 0.6 mL min$^{-1}$ with identical solvents and similar gradient systems. Products were analyzed on an e2695 chromatography work station equipped with a photodiode array detector (PDA) and a QDA-mass detector (Waters, Eschborn, Germany). Products were recorded simultaneously by UV/Vis-detection between 280 and 380 nm (if applicable) and mass detection in a positive ionization mode between $m/z$ 200 and 1200 depending on the substrate and expected product profile. The cone voltage was set at 15 V. Due to the absence of commercial standards, piperine (0.1–100 μM) was used for LC-MS and UV/Vis-based quantification of product formation in the case of all piperamides produced. Kinetic constants for piperine formation were determined in three independent measurements with different enzyme preparation in three technical replicates each.

**Sequence comparisons and cladogram.** Protein sequences included in the cladogram (Fig. 6) were obtained by BLAST searches (Basic Local Alignment Search Tool) using the piperine synthase amino acid sequence as a query against the NCBI non-redundant protein database. Sequences with the highest sequence identities from different species are shown. Accession numbers of BAHD-like crystal structures were obtained from the PDB-database (https://www.rcsb.org/). Protein sequences were aligned, accession numbers listed in the phylogenetic tree, constructed by MegAlign (DNA Star) based on the Clustal V algorithm. For the cladogram, a bootstrap analysis was performed with 1000 replicates. Nucleotide and amino acid sequences were submitted to Genbank (https://www.ncbi.nlm.nih.gov/) and will be released under accession numbers MW354956 (piperine synthase) and MW354957 (piperamide synthase). All protein sequences and complete accession numbers (Fig. 6) are listed as a.fasta file and are included as Supplementary Data 1.

**Statistics and reproducibility.** Statistical analysis of the qRT-PCR was performed using R (Version 3.6.2) as described above. For all statistical analysis, data from at least three independent measurements was used. The exact number of replicates are indicated in individual figure captions and methods.

**Reporting summary.** Further information on research design is available in the Nature Research Reporting Summary linked to this article.

## Data availability

NCBI accession numbers and gene identifiers are listed. Sequence information of piperine synthase (MW354956) and piperamide synthase (MW354957) will be available after the publication of the manuscript. RNA-Seq data were stored in array express and are accessible under the following link: http://www.ebi.ac.uk/arrayexpress/experiments/E-MTAB-9029. All data are accessible in the public repository RADAR (https://www.radar-service.eu). Access is provided using DOI:10.22000/400[51]. Raw data of all files are stored and secured on local IPB-hard drives and IPB-Servers, and can also be provided by the corresponding author (T.V.).

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

## Acknowledgements
The work was generously supported by the German Science Foundation (DFG VO 719/15/1 and 15/2). The authors thank Bernhard Westermann, Jörg Ziegler, as well as Frank Lange (all Leibniz Institute of Plant Biochemistry, Halle) for the critical reading of the manuscript and support of data deposition into RADAR, respectively.

## Author contributions
A.S. performed the major part of the experimental work and evaluated experimental data; B.A. analyzed the data of the RNA-Seq approach; K.M. performed cloning of the constructs and expression of recombinant enzymes; F.S. provided and supervised the growth of the plant material; F.C. contributed to the synthesis of standards; T.V. initiated the project, performed experimental work, and wrote the manuscript with the contribution of all co-authors.

## Funding

## Competing interests
We declare the following competing financial interest: A pending international patent application PCT/EP2020/060165 (issued 9 April 2020) for producing bioactive amides by recombinant piperine synthase and piperamide synthase (illustrated in Fig. 5) has been filed by the Leibniz Institute of Plant Biochemistry (Halle (Saale), Germany), inventors: A.S., B.A., K.M. and T.V. All other authors declare no competing interests.
