## [Peer Review File · Communications Biology]

Reviewers' comments:

Reviewer #1 (Remarks to the Author):

The authors identify and characterize a novel piperine synthase catalyzing the formation of piperine from piperoyl coenzyme A and piperidine using a comparative transcriptomic approach. At the same time, a more promiscuous piperamide synthase is characterized. The work is well conducted, and contains solid classical enzyme kinetics as biochemical evidence and Rt-qPCR analysis of the two identified genes supporting their findings.

The manuscript is written in a clear and concise manner emphasizing the relevance, methodology and potential of the findings. Over all it is a nicely presented story, to which I only have minor suggestions to revision, which in my opinion should be addressed prior to publication.

Abstract, L13-14: Here the authors state the identification of enzymatic formation of piperine from piperoyl coenzyme A and piperic acid, but I understood the identified piperine synthase catalyze formation of piperine from piperoyl coenzyme A and piperidine, not piperic acid. Please clarify this. In general, the introduction of the novel enzymes could be a little clearer in the abstract.

Line 31: "burning sensation", please specify this (I assume oral?).

Line 34: Please delete family.

Line 49: Please delete carbon.

Line 212: Reference 32 is not in superscript.

Line 216: Responsible for volatile formation seems as a very broad term. Please specify.

Line 298: Please add the botanical name in parenthesis after "black pepper" in the methods as well.

Line 315: State that it is de novo assembled, correct?

Line 326: Please add correct version of R (3.x?).

Lines 360-361: Please add rpm after "shaken".

Line 406: Please correct figure reference, it seems to be a reference to the wrong figure.

Discussion: A discussion on how the cellular localization and storage of such high concentrations of piperine is missing. Storage is, however, mentioned in the results section line 171. I suggest a discussion to elaborate on this.

Figure 3: It is not clear to me whether the presented data is from Rt-qPCR or RNA-seq. Please specify.

In addition, it would be an advantage for the reader if the X-axes were named according to tissue type and not only with numbers. A plot showing the piperine content in the respective tissue would be a good complement as well.

Figure S5: Reference to this figure is missing. In line 212, there is a reference to Fig. S6, which is not present in the supplementary material. I assume this should be a reference to Fig. S5.

Supplementary information: I suggest to include the top 10 or 30 expressed transcripts associated with specialized metabolism in which the two identified genes were in top five of (mentioned in line

117) as an additional supplementary figure.

Reviewer #2 (Remarks to the Author):

This manuscript entitled "Piperine synthase from black pepper, *Piper nigrum* » presents results proving the involvement of a BAHD-type acyltransferase enzyme in the biosynthesis of piperine from both substrates, piperic acid and piperoyl CoA. Authors used RNAseq and classical biochemistry approaches to elucidate one more step in the piperine biosynthetic pathway. Piperine is regarded as an option for traditional and recent medicinal applications which makes the elucidation of its biosynthetic pathway important.

This manuscript is well written, concise and with results clearly supporting the conclusions. Authors provide a nice state of the art with all necessary reference to background data. Assumption, hypothesis, and methods are well described. I do not see any major comments to raise, except may be to suggest preparing *P. nigrum* mutants (KO, Knock-Down or surexpressors of the identified piperine synthase) to definitely prove the in planta involvement of the identified BAHD acyltransferase enzyme.

Specific comments:

- 1- In the text at the end of the introduction and at the beginning of the results paragraphs, it seems there is a mixing of the references 27, 28, 29.
- 2- Page 5, line 109: there is a mention to the Fig S2, but this figure is related to a size exclusion chromatography. I suppose Fig S1b is more relevant here.
- 3- To facilitate reading of the Fig 3, I would suggest displaying sample names directly below the abscissa axis.
- 4- Maybe give information of what is loaded in each lane of the PAGE in the inset of Fig 4, and indicate which sample was used for enzyme activity tests.
- 5- How unique is the motif DWGWG? Does the DWGWG motif of piperine and piperamide synthases can be found in some other BAHDs, did you search for it?
- 6- Page 10, line 257: reference 28 seems not appropriate.
- 7- Page 11, Line 268: replace "identity" instead of "identical"
- 8- Page 12, line 327: replace "was" instead of "war"
- 9- A Table 2 is mentioned in the Methods, but no table 2 is available in the manuscript
- 10- Page 14, Line 379: I assume that the molecule name is "3,4-methylenedioxycinnamoyl CoA" instead of "3,4-methylenedioxycinnamoly CoA"
- 11- Page 15, line 406: the reference to Fig 5 seems not appropriate here, I suppose Fig 6 is better.

Response to Referees

Referee 1

The manuscript is written in a clear and concise manner emphasizing the relevance, methodology and potential of the findings.

Thanks, also for your suggestions to further improve the manuscript.

Abstract, L13-14: *Here the authors state the identification of enzymatic formation of piperine from piperoyl coenzyme A and piperic acid, but I understood the identified piperine synthase catalyze formation of piperine from piperoyl coenzyme A and piperidine, not piperic acid. Please clarify this. In general, the introduction of the novel enzymes could be a little clearer in the abstract.*

Of course the referee is right also with respect to a more profound introduction of the enzyme. We corrected the term piperic acid to piperidien and also present the newly identified enzymes more clearly.

Line 31: *“burning sensation”, please specify this (I assume oral?).*

Yes, thank you

Line 34: *Please delete family.*

o.k.

Line 49: *Please delete carbon.*

o.k.

Line 212: *Reference 32 is not in superscript.*

o.k.

Line 216: *Responsible for volatile formation seems as a very broad term. Please specify.*

The referee is right. We explained the term “volatile” in more detail and also addressed the activities of both enzymes in some more detail.

Line 298: *Please add the botanical name in parenthesis after “black pepper” in the methods as well.*

o.k.

Line 315: *State that it is de novo assembled, correct?*

Yes, it is.

Line 326: Please add correct version of R (3.x?).

Corrected 3.6.2

Lines 360-361: Please add rpm after “shaken”.

200 rpm

Line 406: Please correct figure reference, it seems to be a reference to the wrong figure.

Of course, should be Figure 6

Discussion: A discussion on how the cellular localization and storage of such high concentrations of piperine is missing. Storage is, however, mentioned in the results section line 171. I suggest a discussion to elaborate on this.

This reviewer is right, we added a paragraph on the storage of piperine, which for the observed extremely high concentrations (of a very apolar metabolite) appears even more enigmatic than the actual enzyme activity.

Figure 3: It is not clear to me whether the presented data is from Rt-qPCR or RNA-seq. Please specify.

In addition, it would be an advantage for the reader if the X-axes were named according to tissue type and not only with numbers. A plot showing the piperine content in the respective tissue would be a good complement as well.

We agree and added the relevant information, also with respect to the content of piperine in roots. Initially, we were reluctant to sacrifice the limited plant material we are able to grow. Most plants did not contain much root material and we were unable to initially collect roots for RNAseq and also for compound analysis. During the last year we grew additional plants, so the original plants could be harvested for root piperine content. This is now included and also a statement added in the text.

Figure S5: Reference to this figure is missing. In line 212, there is a reference to Fig. S6, which is not present in the supplementary material. I assume this should be a reference to Fig. S5.

Yes, the referee is correct.

Supplementary information: I suggest to include the top 10 or 30 expressed transcripts associated with specialized metabolism in which the two identified genes were in top five of (mentioned in line 117) as an additional supplementary figure.

We included a list for the 30 most abundant transcripts of specialized metabolism. Transcript abundance for piperine synthase is Top 2 (>700 tpm), whereas transcript abundance for piperamide synthase is Top 5 (> 400 tpm), see also Table S1.

Referee 2

*This manuscript is well written, concise and with results clearly supporting the conclusions...except may be to suggest preparing *P. nigrum* mutants (KO, Knock-Down or surexpressors of the identified piperine synthase) to definitely prove the in planta involvement of the identified BAHD acyltransferase enzyme.*

Of course, the referee is right. CRISP/CAS Plants or knockout would be a definite proof. However, black pepper, specifically the fruits, are highly recalcitrant and several efforts to use VIGS were unsuccessful. The berries stop to grow, turn brown, drop of the plants within a few days. Currently, there is no way to create transgenic black pepper, cell cultures are also not available. This would be a project for several years. As one alternative we will try to transiently express all pathway genes in *N. benthamiana*, as soon as the remaining steps towards the formation piperine are available (a report on the identification of an enzymatically active CYP719 coexpressed with these BAHDs from fruits will be prepared in due course).

Specific comments:

1- In the text at the end of the introduction and at the beginning of the results paragraphs, it seems there is a mixing of the references 27, 28, 29.

Yes, sorry for the inconvenience, this has been corrected.

2- Page 5, line 109: there is a mention to the Fig S2, but this figure is related to a size exclusion chromatography. I suppose Fig S1b is more relevant here.

Yes, thank you this is also correct

3- To facilitate reading of the Fig 3, I would suggest displaying sample names directly below the abscissa axis.

Consistent also with the suggestion by Reviewer 1, we made this change, hope it is clearer now.

4- *Maybe give information of what is loaded in each lane of the PAGE in the inset of Fig 4, and indicate which sample was used for enzyme activity tests.*

Thank you, we combined the purified samples to have sufficient material for all enzyme and specificity tests. As indicated we performed several rounds of purification to have sufficient material, specifically BAHD2 was expressed only in low amounts. We are aware that some of the purified material could be aggregating with somewhat reduced activity. However, we currently do not have the possibility to prevent this aggregate formation, which make take place at every step of purification.

5- *How unique is the motif DWGWG? Does the DWGWG motif of piperine and piperamide synthases can be found in some other BAHDs, did you search for it?*

It is truly unique, we did not find any other BAHD-sequence with this motif. Since it is present in both, piperine synthase and piperamide synthase, it is not responsible

for the observed differences in substrate specificity. It seems also localized far outside of the active center, judged from structures of crystallized BAHDs. We have started crystallisation trials to elucidate its relevance and once suitable crystals are obtained we are confident that site directed mutagenesis will then provide detailed specific relevance of individual amino acids for activity and stability. A comment on this aspect is included in the manuscript.

6- Page 10, line 257: reference 28 seems not appropriate.

In fact, should be reference 29.

7- Page 11, Line 268: replace “identity” instead of “identical”

o.k.

8- Page 12, line 327: replace “was” instead of “war”

o.k.

9- A Table 2 is mentioned in the Methods, but no table 2 is available in the manuscript

Thank you, well, this Table is now included in the Supplemental Files, we did not want to release the primer sequences in the first draft. Now they are included, sorry for the initial problems, we should have mentioned that.

10- Page 14, Line 379: I assume that the molecule name is “3,4-methylenedioxycinnamoyl CoA” instead of “3,4-methylenedioxycinnamoly CoA”

Yes, of course, thanks

11- Page 15, line 406: the reference to Fig 5 seems not appropriate here, I suppose Fig 6 is better.

Yes, of course, it should be Figure 6.

REVIEWERS' COMMENTS:

Reviewer #1 (Remarks to the Author):

I thank the authors for the revised manuscript, which I enjoyed reading. I find the additions to figure 3 including a more detailed description, and the addition of table S1 very useful in the understanding of the work. Moreover, the expansion of the results and following discussion fully meets the concerns addressed in my previous comments.

Only two minor comments regarding figures/tables. In line 242, the authors refer to Fig. S1 with regard to expression of BAHD-like enzymes, but how can I see that from the mentioned figure? I can see from the figure that the transcripts in the fruit stages differentiate from those in leaves and flowers based on the PCA, but I am not sure how that gives me information about the BAHD-like enzymes. Perhaps I just do not get the point, and in that case, I apologize if it is a naïve comment. Finally, in line 370 list of primers referred to in Table S1 must be Table S2 now.

Reviewer #2 (Remarks to the Author):

This revision of the manuscript "Piperine Synthase from Black Pepper, *Piper nigrum*" presents clear results proving the involvement of a BAHD-type acyltransferase enzyme in the biosynthesis of piperine.

This new version of the manuscript is still well written, and it carefully answers all comments raised by reviewers.

I have two more minor remarks: 1) Please clearly describe the figure 3C in its related legend ; 2) line 242, page 10, I would replace "fairly high expressed" by "fairly highly expressed"

Response to Referees – 11-02-2021

Reviewer #1

I thank the authors for the revised manuscript, which I enjoyed reading. I find the additions to figure 3 including a more detailed description, and the addition of table S1 very useful in the understanding of the work. Moreover, the expansion of the results and following discussion fully meets the concerns addressed in my previous comments.

- Thank you!

Only two minor comments regarding figures/tables. In line 242, the authors refer to Fig. S1 with regard to expression of BAHD-like enzymes, but how can I see that from the mentioned figure? I can see from the figure that the transcripts in the fruit stages differentiate from those in leaves and flowers based on the PCA, but I am not sure how that gives me information about the BAHD-like enzymes. Perhaps I just do not get the point, and in that case, I apologize if it is a naïve comment.

- Of course, you are absolutely right, this was our mistake - it should read: Table S1 and not Figure S1. The enzyme is listed among the 30 most abundant specialized metabolite transcripts. Thank you for pointing this out.

Finally, in line 370 list of primers referred to in Table S1 must be Table S2 now.

- Yes, since a new Table S1 is included it should be and is Table S2.

Reviewer #2 (Remarks to the Author):

This revision of the manuscript “Piperine Synthase from Black Pepper, Piper nigrum” presents clear results proving the involvement of a BAHD-type acyltransferase enzyme in the biosynthesis of piperine. This new version of the manuscript is still well written, and it carefully answers all comments raised by reviewers.

Thank you

I have two more minor remarks: 1) Please clearly describe the figure 3C in its related legend ; 2) line 242, page 10, I would replace “fairly high expressed” by “fairly highly expressed”

- Sorry, yes, we missed that. A detailed legend has been added in Figure 3C and we changed the term in line 242 to “fairly highly expressed”.